# Adenosine-Mimicking Derivatives of 3-Aminopyrazine-2-Carboxamide: Towards Inhibitors of Prolyl-tRNA Synthetase with Antimycobacterial Activity

**DOI:** 10.3390/biom12111561

**Published:** 2022-10-26

**Authors:** Vinod Sukanth Kumar Pallabothula, Marek Kerda, Martin Juhás, Ondřej Janďourek, Klára Konečná, Pavel Bárta, Pavla Paterová, Jan Zitko

**Affiliations:** 1Faculty of Pharmacy in Hradec Králové, Charles University, Ak. Heyrovského 1203/8, 500 05 Hradec Králové, Czech Republic; 2Department of Clinical Microbiology, University Hospital Hradec Králové, Sokolská 581, 500 05 Hradec Králové, Czech Republic

**Keywords:** 3-aminopyrazinamide, antibacterial, antimycobacterial, homology model, molecular dynamics, prolyl-tRNA synthetase, repurposing, tuberculosis

## Abstract

Multidrug-resistant tuberculosis (MDR-TB) poses a significant threat to mankind and as such earned its place on the WHO list of priority pathogens. New antimycobacterials with a mechanism of action different to currently used agents are highly required. This study presents the design, synthesis, and biological evaluation of 3-acylaminopyrazine-2-carboxamides derived from a previously reported inhibitor of human prolyl-tRNA synthetase. Compounds were evaluated in vitro against various strains of mycobacteria, pathogenic bacteria, and fungi of clinical significance. In general, high activity against mycobacteria was noted, while the antibacterial and antifungal activity was minimal. The most active compounds were 4’-substituted 3-(benzamido)pyrazine-2-carboxamides, exerting MIC (Minimum Inhibitory Concentration) from 1.95 to 31.25 µg/mL. Detailed structure–activity relationships were established and rationalized in silico with regard to mycobacterial ProRS as a probable target. The active compounds preserved their activity even against multidrug-resistant strains of *Mycobacterium tuberculosis*. At the same time, they were non-cytotoxic against HepG2 human hepatocellular carcinoma cells. This project is the first step in the successful repurposing of inhibitors of human ProRS to inhibitors of mycobacterial ProRS with antimycobacterial activity.

## 1. Introduction

The antimicrobial resistance (AMR) phenomenon is considered one of the highest-ranked threats to global human health. If no proper measures are taken today, it is estimated that by 2050, the deaths due to AMR infections will reach 10 million deaths/year (versus 8.2 million deaths/year due to all types of cancer) [1]. The most clinically significant bacteria with AMR include methicillin-resistant Staphylococcus aureus (MRSA) [2], coagulase-negative staphylococci (CoNS) [3], and in general, Gram negative bacteria [4]. All these pathogens fall into the WHO priority list of pathogens to combat, along with multidrug-resistant tuberculosis (MDR-TB) [4]. In 2020, 10 million people were estimated to develop active TB, and 1.5 million died (including 0.2 million among HIV positives). Globally in 2019, 3.3% of new TB cases and 18% of previously treated TB cases were multidrug-resistant (MDR) or at least rifampicin-resistant [5].

Aminoacyl-tRNA synthetases (aaRS) are a family of essential enzymes that ligate appropriate amino acids to their corresponding tRNAs. A two-step catalytic reaction involves the formation of an enzyme-bound aminoacyl-adenylate (AMP-aa), followed by the formation of an aminoacylated tRNA by transferring the amino acid to the corresponding tRNA. This catalytic reaction of aaRSs plays a pivotal role in proteosynthesis, which is essential for the growth and survival of all cells. The aaRS have been validated as perspective antimicrobial targets [6], as the differences in divergence between prokaryotic and eukaryotic cytoplasmic aaRS offer the possibility to design selective inhibitors [7]. Currently, there are three inhibitors of various aaRS in clinical practice as antimicrobial agents: mupirocin as a competitive inhibitor of bacterial isoleucyl-tRNA synthetase (IleRS), tavaborole as an irreversible inhibitor of fungal leucyl-tRNA synthetase (LeuRS), and halofuginone as a non-competitive inhibitor of protozoal prolyl-tRNA synthetase (ProRS) [8]. In the antimycobacterial research, the potential of aaRS inhibitors has been rather underestimated. The most studied mycobacterial aaRS is LeuRS with at least four structural types of inhibitors, followed by TyrRS and AspRS. Inhibitors of mycobacterial MetRS, LysRS, and PheRS were addressed only sporadically [8]. In this study, we have focused on so far unexploited mycobacterial ProRS.

The design of our compounds is based on the confirmed ATP-competitive inhibitor of human ProRS (hsProRS) with 3-aminopyrazine-2-carboxamide scaffold (further referenced as Adachi ligand) [9]. This inhibitor binds to the ATP site of the enzyme, mimicking the interactions of the adenine core. We compared the binding mode of the Adachi inhibitor (pdb id: 5VAD) and adenosine (pdb id: 2J3L). Three analogous pharmacophore features, hydrogen bonds, can be observed in the overlay of these two crystallographic structures (Figure 1). This led us to the presumption that these hydrogen bonds are important or even necessary for binding to the enzyme.

In our previous study, we demonstrated that similar 3-aminopyrazinamide derivatives with simplified substituents (Figure 2, in comparison to the Adachi ligand) could also bind to hsProRS (as demonstrated by thermal shift assay and crystallographic structures) [10], although with weaker affinity. The weaker affinity could have been due to the missing carbonyl oxygen. In our previous series, compounds with 2’-substitution on the benzene ring had affinity to hsProRS [10], but were devoid of significant antimycobacterial activity [11]. On the contrary, compounds with 4’-substituent did not bind to hsProRS [10] but possessed moderate in vitro growth inhibition activity against mycobacteria [11]. In this paper, we aimed to obtain new target compounds with affinity to mycobacterial ProRS and in vitro antimycobacterial activity, but devoid of significant toxicity on human cells. The compounds were designed by simple formal addition of the carbonyl oxygen to our previous structures, and by exploring the SAR in regard to the substitution on the benzene ring. To confirm the importance of the carboxamidic group at C-2 for binding, we also prepared derivatives with substituted carboxamide and cyclized derivatives (Figure 2).

## 2. Materials and Methods

### 2.1. General

All chemicals (unless stated otherwise) were purchased from Sigma-Aldrich (Schnelldorf, Germany). The reaction progress and the purity of the final compounds were checked using Merck Silica 60 F_254_ TLC plates (Merck, Darmstadt, Germany). Purification of synthesized compounds was performed on automated flash chromatograph puriFlash XS420+ (Interchim, Montluçon, France) using original columns (silica, 30 µm). The mixture of ethyl acetate and hexane was used as a mobile phase with elution gradient 0–100% ethyl acetate (EtOAc) to hexane (Hex), and detection was performed by UV-VIS detector at a wavelength of 254 nm and 280 nm. NMR spectra were recorded on Varian VNMR S500 (Varian, Palo Alto, CA, USA) at 500 MHz for ^1^H and 125 MHz for ^13^C or using Jeol JNM-ECG600 at 600 MHz for ^1^H and 151 MHz for ^13^C. The spectra were recorded in DMSO-*d_6_* at ambient temperature (unless stated otherwise). The chemical shifts as δ values in ppm are indirectly referenced to tetramethylsilane (TMS) via the solvent signal. Nicolet 6700 spectrometer (Thermo Scientific, Waltham, MA, USA) was used to measure IR spectra using ATR-Ge method. Chemical composition was recorded by an elemental analysis performed on Vario MICRO cube Element Analyzer (Elementar Analysensysteme, Hanau, Germany) with values given as percentages. Melting points of the synthesized compounds were measured in an open capillary on Stuart SMP30 melting point apparatus (Bibby Scientific Limited, Staffordshire, UK) and are uncorrected. Yields are given as percentages and refer to the amount of chromatographically pure product obtained after all the purification steps. Log P values were calculated using ChemDraw 20.0 (PerkinElmer Informatics, Waltham, MA, USA).

### 2.2. Synthesis and Purification—General Procedures

#### 2.2.1. Acylation of Methyl 3-Aminopyrazine-2-Carboxylate

*Procedure A:* 4 mmol of starting material was dispersed in 20 mL of dichloromethane (DCM) under an inert atmosphere (argon). After stirring for 5 min, 1.5 eq of anhydrous pyridine was added and stirred for another 5 min; 1.2 eq of acyl chloride was added dropwise to the reaction mixture at continuous stirring. The reaction continued for 48 h at room temperature. The reaction progress was checked on TLC using 1:1 Hex; EtOAc. Since the nucleophilicity of the starting material is low, a partial presence of the starting material was observed on the TLC even after the reaction was stirred for more than 48 h. Once the reaction was complete, the solvent was evaporated under vacuo. The crude reaction mixture was purified multiple times on automated flash chromatography due to the presence of unreacted starting material with gradient elution using Hex/EtOAc (from 0% of EtOAc to 100%) and detected at absorption wavelength of 254 and 280 nm.

*Procedure B1:* 4 mmol of starting material was dispersed in 20 mL of acetonitrile (ACN) and stirred at room temperature for 5 min, followed by the addition of 2.5 eq of anhydrous pyridine. The reaction mixture was stirred for another 5 min and 2.2 eq of acyl chloride was added dropwise. The reaction temperature was increased to 70 °C and refluxed for 24 h. Progress was checked on TLC with the same conditions as *Procedure A.* Once the reaction was completed with full consumption of the starting material, the solvent was evaporated under vacuo. The crude reaction mixture was purified on automated flash chromatography with gradient elution using Hex/EtOAc (from 0% EtOAc to 100%) and detected at absorption wavelengths of 254 and 280 nm. Both the diacylated and monoacylated products were separated.

*Procedure B2*: The conversion of diacylated to monoacylated product was achieved by a reduction reaction [12]. To 1 mmol of diacylated product in 10 mL of tetrahydrofuran (THF), 3 mL of isopropanol was added as a catalyst followed by 1 mL of 1 M hydrazine hydrate in THF. The reaction mixture was refluxed for 1 h. Once the reaction was finished, the solvent was evaporated and the contents were dispersed in acidic water (few drops of H_2_SO_4_) and extracted to EtOAc. The yield of this reduction to monoacylated intermediate was 95 to 97%.

#### 2.2.2. Ammonolysis of Methyl Ester Moiety (Compounds **1–24**)

*Procedure C*: 1 mmol of the monoacylated intermediate was added to an excess of 2 M ammonia in anhydrous ethanol. The reaction mixture was stirred continuously at room temperature for 24 h. The solvent and the unreacted ammonia were evaporated, and the final white to beige solid compound was recovered with 90 to 99% yield. 

#### 2.2.3. Synthesis of 3-(4-hydroxybenzamido)pyrazine-2-carboxamide (Compound **22**) and 3-(2-hydroxybenzamido)pyrazine-2-carboxamide (Compound **23**):

Protection of the hydroxy group was necessary before the acylation step. Hydroxybenzoic acid was dispersed in a mixture of toluene and tetrahydrofuran in the 4:1 ratio. Pyridine (4 eq) was added and the reaction mixture was cooled to 0 °C. After that, acetic anhydride (1.1 eq) was added dropwise. The reaction was left to heat to rt and stirred overnight. Reaction mixture was evaporated to dryness and washed with acidic water achieving crude white solid (100% yield). For acylation of methyl 3-aminopyrazine-2-carboxylate, *Procedure B* was used with accent to anhydrous conditions. After the reaction and purification were completed, the acetyl groups of acetoxy derivatives were cleaved during ammonolysis (*Procedure C*). 

#### 2.2.4. Synthesis of 3-(4-aminobenzamido)pyrazine-2-carboxamide (Compound **24**): 

The intended compound was achieved by reducing 3-(4-nitrobenzamido)pyrazine-2-carboxamide (compound **21**). The best conditions for the reduction were achieved in ethanol with acetic acid and an excess of zinc. The reaction was heated to 50 °C for 24 h followed by filtration and were evaporated to dryness. The crude product was purified using flash chromatography (gradient elution of Hex/EtOAc/MeOH) and recrystallization in ethanol. 

#### 2.2.5. Synthesis of Secondary Carboxamide Derivatives (Compounds **33–35**):

A total of 10 mmol of starting material was dispersed in methanol and an excess amount of methyl amine was added. The reaction mixture was stirred continuously at room temperature for 24 h. The solvent and the unreacted methyl amine were evaporated under vacuo to obtain 3-amino-*N*-methylpyrazine-2-carboxamide. The reaction continued in the same way as in *Procedure B*. White solids with 75 to 85% yields were obtained.

#### 2.2.6. Synthesis of Tertiary Carboxamide Derivatives (Compounds **36** and **37**):

The acylated intermediates were prepared with *Procedure B.* Then, 2 mmol of the intermediate was obtained, dispersed in methanol, and excess dimethylamine was added. After completion, the solvent and the unreacted dimethylamine were evaporated under vacuo. The solid product was obtained with 90 to 95% yield. 

#### 2.2.7. Cyclization to Pteridine Derivatives (Compounds **39**–**43**):

A total of 1 mmol of the compound was dissolved in 10 mL of 0.5 m potassium hydroxide in water and 2 mL of DMSO was added as a cosolvent (the temperature increased to 60 °C to achieve complete dissolution) and stirred continuously for 1 h. Once the reaction was completed and cooled to rt, acidic water was poured into the reaction. A white precipitate was formed and filtered under vacuo. The crude product was washed several times with distilled water and dried in a hot air oven for about 2 h.

### 2.3. In Silico Simulations

#### 2.3.1. Software

In silico calculations were performed in Molecular Operating Environment (MOE) 2022.09 (Chemical Computing Group Inc., Montreal, QC, Canada) under Amber10:EHT forcefield if not otherwise stated. Molecular dynamics simulation was run using NAMD (University of Illinois at Urbana-Champaign, IL, USA) [13]. Stages 1–5 were simulated using NAMD 2.10. The production phase 6 was calculated using NAMD 3 alpha 9 utilizing the CUDA GPU acceleration. Trajectory analysis was performed using VMD (Visual Molecular Dynamics, version 1.9.4a53, University of Illinois at Urbana-Champaign, IL, USA) [14] after aligning the trajectory on the backbone atoms (C, Cα, N) of the central domain. The central domain was defined as ‘residues (415 to 470) or residues (20 to 234)’. Figures were generated using MOE or MS Excel (Redmond, Washington, United States).

#### 2.3.2. Molecular Docking

The ligands for docking were generated from SMILES, using the MOE built-in function to predict the dominant protonation state at pH 7. 3D coordinates were minimized until RMS gradient 0.01 kcal.mol^−1^.Å^−1^.

Three-dimensional coordinates of mycobacterial prolyl-tRNA synthetase (mtProRS) were downloaded from the AlphaFold database (UniProt ID: P9WFT9). The model was overlaid with experimental coordinates of ProRS from *Enterococcus faecalis* (pdb id: 2J3L), using the alignment used to generate the homology model by AlphaFold. We used the overlay of the two structures to manipulate proline (Pro) and adenosine as a substrate and a substrate-like compound, respectively, into the homology model. The rotamer of Thr111 sidechain was changed to match the conformation of the corresponding Thr in the template (necessary for the interaction with Pro substrate). Additionally, we kept the water molecule forming the water bridge between adenosine N-3 and Ser461 as seen in efProRS (pdb id: 2J3L). The resulting system (mtProRS homology model with Pro, adenosine, one molecule of water, and manipulated Thr111) was prepared by MOE QuickPrep functionality with default settings, which included corrections of structural errors, the addition of hydrogens, calculation of partial charges, 3D optimization of protonation/tautomeric states and *H*-bond network (Protonate3D) and a restrained minimization (to RMS gradient of 0.01 kcal.mol^−1^. Å^−1^). 

Proline was set as a part of the receptor. The pocket was defined as a set of residues with at least one atom within 4.5 Å from the overlayed ligand (adenosine). Parameters of the MOE docking protocol: Docking stage–Placement: Triangle Matcher; score: London dG; retain 50 poses: Refinement stage–Rigid receptor or induced fit (default settings with tethering of pocket side chains); score: GBVI/WSA dG; retain 5 poses; Ligand conformation–Rotate bonds.

#### 2.3.3. Molecular Dynamics

The system for MD simulation was constructed from the selected docked pose (expected interactions and best by docking score) of compound **15**. Input for the MD simulation was prepared in MOE, applying the parameters from Amber10:EHT forcefield. The system was solvated using TIP3P waters in a 10 Å margin periodic boundaries box, neutralized, and buffered using NaCl (c = 0.15 M). In the simulation, non-bonded Van der Waals interactions were truncated (switching distance 10, cutoff distance 12). Long-range electrostatics were treated using the Particle Mesh Ewald (PME). Bonds to hydrogen atoms were constrained using the ShakeH algorithm (with a default convergence criterion of 1.0 × 10^−8^). The temperature was controlled by Langevin dynamics, and the pressure was treated using the Nosé–Hoover–Langevin piston pressure control, both as implemented in NAMD. The time step was set to 2 fs and the coordinates were recorded each 10 ps.

MD Protocol (temperature T in kelvins, pressure P in bar, r is a restrain to heavy atoms as defined in MOE, see below *).

Stage 1: Restrained minimization {ps = 10 T = 0 r = 0.5};

Stage 2: Unrestrained minimization {ps = 10 T = 0};

Stage 3: Heating with gradually released restraints {ps = 180 T = (10,300) r = (0.5,10)};

Stage 4: NVT equilibration {ps = 200 T = 300};

Stage 5: NPT equilibration {ps = 600 T = 300 P = 1};

Stage 6: NPT production {ps = 50,000 T = 300 P = 1};

* Restraint: Heavy atom tether restraint in Å. A value of 0 means that heavy atoms are fixed, while, for example, 1 means that a restraint force constant that produces a 1 Å radial standard deviation from the reference position will be applied. If no restraint gradient is defined, this is a constant restraint applied during the entire MD segment.

## 3. Results and Discussion

### 3.1. Chemistry

We tried two different procedures for the acylation of methyl 3-aminopyrazine-2-carboxylate (starting material) with R-substituted benzoyl chlorides, alkanoyl chlorides and cycloalkanoyl chlorides (Figure 1). Direct acylation of 3-aminopyrazine-2-carboxamide failed due to the significantly low nucleophilicity of the amine. Methyl 3-aminopyrazine-2-carboxylate has slightly higher reactivity.

*Procedure A* involves argon as an inert atmosphere, pyridine as a base, and DCM as a solvent and is carried out at room temperature. Such conditions lead to yields ranging from 40 to 60% with a predominantly monoacylated product. Due to the partial consumption of the starting material and similar retention factors between the starting material and the desired product, it was tough to isolate pure compounds using chromatographic techniques. We commonly used a mixture of hexane and ethyl acetate.

*Procedure B* (B1 and B2) aims to increase the reactivity by using higher temperatures up to 70 °C, pyridine as a base and acetonitrile solvent. The reaction produces higher yields of diacylated products and lower yields of monoacylated products. Although diacylation was initially undesired, conversion to a monoacylated product can be achieved by a simple reduction using hydrazine hydrate. Synthesizing diacylated products simplifies the purification steps using chromatographic techniques because the difference in retention factors is much more significant than in *Procedure A.* The overall yields of the final monoacylated products achieved from *Procedure B* ranged from 90 to 95% after all purification steps.

Final compounds were synthesized with *procedure C* by simple ammonolysis of substituted methyl 3-benzamidopyrazine-2-carboxylates with an excess of 2 M ammonia in ethanol. The reaction finishes in 24 h by producing white or beige-colored precipitate suspended in the reaction mixture. The evaporation of solvent and ammonia under reduced pressure gives 90 to 99% yield of the final products.

Compounds with polar substituents on the benzene ring needed a modified synthetic approach. The Hydroxyl group in hydroxy derivatives had to be protected before the acylation step. *Procedure B* was used with an accent to anhydrous conditions. Amino derivatives were synthesized by a reduction in nitro derivatives.

As a complementary SAR study, we synthesized some derivatives of the secondary and tertiary carboxamides of their respective final compounds according to Figure 2. For compounds **33**–**35**, the starting material firstly underwent aminolysis from methyl ester to methyl carboxamide using methyl amine (Procedure i). After that, the synthesis continued with acylation *Procedure B* (ii) described above. Compounds **36**–**38** were synthesized using acylation *Procedure B* (ii) followed by aminolysis with dimethylamine (procedure iii). 

The series was further extended by synthesis of pteridine derivatives from compounds **1**, **4**, **12**, **15** and **18** using cyclization with potassium hydroxide solution according to Figure 1. A white precipitate was observed after the acidification of the reaction mixture to pH 5. The precipitate was filtered and dried in an air oven at 90 °C for 2 h.

### 3.2. Analytical Description

All final products were characterized by melting point, ^1^H and ^13^C NMR spectra measured in hexadeuterodimethyl sulfoxide (DMSO-*d_6_*) at ambient temperature (unless otherwise stated), IR spectroscopy, elementary analysis, and mass spectrometry. Due to the ambiguous results of the pteridine derivatives (compounds **39**–**43**) from elementary analysis, the compounds were checked with HPLC analysis. The analytical data of all final compounds and the intermediates correspond to the proposed structures. 

Description of typical ^1^H NMR spectra: Carboxamide protons (CONH_2_) with two broad singlets were observed at 8.7 to 8.3 and 8.5 to 7.8, amidic proton (NHCO) at 13.0 to 11.5 ppm with a sharp singlet, and two doublets of pyrazine protons at 8.8 to 8.2 ppm. For compounds 39–43, the solvent for the measurements was heated to 40 °C. Amidic protons (NHCO) were observed from 13.5 to 12.9 ppm, two sharp doublets from pyrazine hydrogens at 8.5 to 9.3 ppm. Characteristic NMR spectra of the synthesized compounds are presented in the Appendix A.

Description of typical IR spectra: Strong carbonyl stretching of carboxamide signals were observed from 1665 to 1716 cm^−1^, sharp signal of NH stretching in the ranges of 3430 to 3480 cm^−1^, broad C–H stretching of aromatic groups at 3200 to 3100 cm^−1^, while the CH signals in the alkyl group at 2950 to 2800 cm^−1^, phenyl C = C stretching in the range of 1620 to 1400 cm^−1^. The C–O stretching in ester intermediates was observed at approximately 1117 cm^−1^.

### 3.3. Biological Evaluation

#### 3.3.1. In Vitro Antimycobacterial Evaluation

All the prepared final compounds were diluted to final concentrations of 500–250–125–62.5–31.25–15.62–7.81–3.91–1.95 µg/mL using a serial dilution technique. *Mtb* H37Ra (avirulent strain), *M. smegmatis,* and *M. aurum* were used for initial screening for in vitro whole-cell antimycobacterial activity using a modified Microplate Alamar Blue Assay (MABA) [15]. Results of antimycobacterial activity were measured as MIC in µg/mL and compared with standards isoniazid (INH), rifampicin (RIF), and ciprofloxacin (CIP)-see Table 1, Table 2, Table 3, Table 4. If not otherwise stated, the term activity refers to the activity against Mtb H37Ra from the primary screening.

In total, 56 compounds were screened for antimycobacterial activity. The highest activity was observed in 3-benzamidopyrazine-2-carboxamides **1**–**24**. Based on the results presented in Table 1, a further substitution of the benzamide moiety was favorable, especially with a lipophilic substituent in position 4 of the benzene ring. Interestingly, none of the derivatives substituted in position 2 or 3 had any significant activity, indicating a clear structural requirement for compounds active against Mtb. The most active derivatives bore a halogen (**18** R = 4-Br, **15** R = 4-Cl, **12** R = 4-F), closely followed by an aliphatic substituent (**4** R = 4-Me, **6** R = 4-*t*Bu, compounds listed in activity-descending order). The typical MIC values against Mtb ranged from 1.95 to 31.25 μg/mL. Seemingly, the high antimycobacterial activity is not tied only to the electron-withdrawing groups (EWG > EDG, e.g., **15** vs. **4**) but its size also plays a role (bulkier substituents are more preferred e.g., **4** vs. **12**). Derivative **18** (R = 4-Br) fulfilled both lipophilicity and bulkiness criteria and was the most active among all tested compounds against Mtb (MIC = 1.95 μg/mL). Higher activity of the derivatives containing larger halogen atoms (Cl or Br) could also mean a formation of a halogen bond to the receptor, as further investigated in silico. Surprisingly, compound **5** with Et in position 4 had no activity compared to its smaller or higher homologues **4** (R = 4-Me) and **6** (R = 4-*t*Bu). We hypothesize that the higher flexibility of the Et group over Me or *t*Bu could be the reason. However, the conclusion would require more detailed evaluations. Derivative **15** with a 4-Cl substitution also showed high activity against *M. smegmatis*, missing in the 4-Br derivative **18** or 4-F **12**, that were selective only against Mtb.

Elongating the distance between the 3-carbonylamino group of the central pyrazinamide core and the benzene substituent with an insertion of a methylene or ethylene linker (derivatives **25** and **26**) led to inactive compounds. Replacing the aromatic core with the aliphatic or alicyclic moieties (**27–32)** also led to inactive compounds, with the exception of the adamantoyl substituted compound **28**, which showed mediocre activity against Mtb (MIC = 31.25 ug/mL) but not against other strains of mycobacteria (Table 2).

All the tested methyl ester intermediates of the final compounds containing ester moiety at C-2 instead of the carboxamide were completely inactive against tested species (results in Appendix A). This clearly showed that the NH_2_ hydrogen bond donor in the C-2 carboxamide moiety plays a pivotal role in the interaction with the supposed cellular target. To test this hypothesis further, we prepared *N*-methyl- and *N*,*N*-dimethylcarboxamide derivatives (**33**–**37**) of selected active compounds, thus cancelling one or both hydrogen bonds to the enzyme (mtProRS) (Table 3). As expected, the *N*,*N*-dimethyl derivatives (**36**, **37**) were inactive. In the optics of interaction with the supposed target mtProRS, this would be explained by the inability to form the crucial H-bond to Ala154 backbone. Surprisingly, the *N*-monomethyl derivatives **33**–**35** had low or no activity, and this cannot be explained by the incompatibility with the expected binding mode, as one carboxamidic hydrogen is still compliant with the binding mode to mtProRS. 

Constraining the conformation of the most active compounds by cyclization (compounds **39**–**43**) significantly decreased the activity in comparison with non-cyclized precursors (Table 4). The cyclization traps the C-2 carboxamide in conformation non-compliant with the expected binding mode with NH atom of the C-2 carboxamide pointing ‘down’ towards the C-3 carbonyl instead the normal ‘NH up’ conformation, which is driven by intramolecular H-bond formation and proven by crystallographic structures of previously published derivatives of 3-aminopyrazine-2-carboxamide in hsProRS [10].

To sum up, our experiments confirmed the importance of both hydrogens of the unsubstituted carboxamide at C-2 for the antimycobacterial activity as well as the importance of the amino moiety of the C-2 carboxamide pointing upwards. In the derivatives with benzene moiety at the C-3 substituent, we observed the preference for the 4’-substitution over 2’-substitution, which is consistent with structure–activity relationships in the 3-benzylamino-2-pyrazine-2-carboxamide series (one of the starting points for the design of this study) [11].

Described structure–activity relationships, i.e., 4’-substitution in the benzamide substituent and the importance of the carboxamide group, were further investigated in silico as mentioned below.

#### 3.3.2. In Vitro Antimycobacterial Evaluation on Drug-Sensitive and Multidrug-Resistant Strains of *M. Tuberculosis*

The selected most active compounds from the primary screening of antimycobacterial activity were further subjected to testing against the virulent reference strain *M. tuberculosis* H37Rv and two multidrug-resistant clinical isolates of Mtb (Table 5). The clinical isolates were resistant to streptomycin and almost all first-line antituberculars, namely isoniazid, rifampicin, and pyrazinamide, and were only sensitive to ethambutol. For the complete resistance profile of the isolates, see Appendix A. Compounds **4**, **12,** and **15** proved to have significant activity not only on the virulent strain Mtb H37Rv, but also on both MDR strains. Given the commonly perceived error of the broth dilution method (two steps on the binary dilution scale), it can be stated that the compounds preserved their activity on MDR strains. This indicates that our compounds act through a novel molecular target, not shared with the commonly used antimycobacterial drugs. Specifically, the activity against strains resistant to pyrazinamide indicates that, although structurally derived from pyrazinamide, our compounds do not share its mechanism of action.

#### 3.3.3. In Vitro Antibacterial and Antifungal Testing

For antibacterial and antifungal evaluation, microplate broth dilution was performed according to the slightly modified methodology of The European Committee on Antimicrobial Susceptibility Testing (EUCAST) in correspondence with our earlier publication [16]. Various strains of pathogenic bacterial strains and fungi were used to determine the inhibitory activity of the prepared compounds. None of the final compounds exhibited significant activity on any of the tested strains, see Appendix A). 

#### 3.3.4. In Vitro Cytotoxic Studies

The pharmacology intervention in the treatment of tuberculosis is always multi-component. Therefore, newly developed antimycobacterial compounds usually enter the clinical evaluations in combinations with currently used agents. Since many commonly used antimycobacterial drugs are hepatotoxic, we screened our compounds for in vitro cytotoxicity using a hepatocellular carcinoma cell line model (HepG2). We also had in mind that the design of our compounds started from the Adachi ligand, a confirmed inhibitor of hsProRS [9], which, by its nature, should be cytotoxic on human cells. We used a standard colorimetric method based on the metabolic reduction of tetrazolium salt to determine the viability of the HepG2 cells. The results are presented as the half-maximum inhibitory concentration (IC_50_), a concentration of the tested compound which reduces the viability to 50% compared to untreated control.

Compound **38** (Adachi, hsProRS inhibitor) was confirmed to have a significant cytotoxic activity (IC_50_ = 19.6 µM). In this relation, most of our compounds had IC_50_ > 100 µM and were therefore considered non-cytotoxic. Compounds **11, 14, 17,** and **18** were limited by low solubility in the testing medium. Complete results of cytotoxicity evaluation are presented in Table 1, Table 2, Table 3, Table 4. A complete description of the method is presented in the Appendix A. 

### 3.4. In Silico Simulations 

#### 3.4.1. Docking to Homology Model of Mycobacterial ProRS

To assess the probability whether our 3-benzamidopyrazine-2-carboxamides with antimycobacterial activity could act via the mechanism inferred from the design of the compounds, that is, the interaction with mycobacterial ProRS, we performed molecular docking into ProRS of *M. tuberculosis* (mtProRS), followed by unbiased molecular dynamics simulations of the resulting representative complexes.

The crystallographic structure of mtProRS (UniProt ID: P9WFT9) is not available; therefore, we used a homology model constructed by AlphaFold-an artificial intelligence system developed by DeepMind [17]. The main template used to build the model was a bacterial ProRS from *Enterococcus faecalis* (pdb id: 2J3L). Therefore, we took advantage of the template-target protein alignment determined by AlphaFold and used the overlay of the two structures to manipulate proline (Pro) and adenosine as a substrate and a substrate-like compound, respectively, into the homology model. Additionally, we kept the water molecule forming the water bridge between adenosine N-3 and Ser461 as seen in EfProRS (pdb id: 2J3L). This water bridge often occurs also in other crystallographic structures of ProRSs and participates in the interactions. We confirmed the importance of this water molecule by Solvent Analysis application using 3D-RISM model [18]. For this reason, we considered this water molecule crystallographic and kept it in the receptor for docking. The pocket for docking was determined by the position of the adenosine and our compounds were docked alongside the Pro substrate, because the binding of similar 3-aminopyrazinamide derivatives to hsProRS was Pro-dependent (compounds did not bind without prior bound Pro) [10].

Firstly, we docked compounds **1**–**24** (compounds with unsubstituted carboxamide at C-2 and benzamido substituent at C-3) as members of the only series which brought significant antimycobacterial activity. The obtained representative binding mode was fully consistent with the crystallographic binding mode described for 3-aminopyrazinamide derivatives in hsProRS, [10] with the pyrazine-2-carboxamide moiety forming a donor-acceptor dyad of HBs to single amino acid sidechain (Thr1164 in hsProRS, corresponding to Ala154 in mtProRS). The intramolecular HB (IHB) between 3-amino (donor) and carbonyl oxygen from the 2-carboxamide moiety (acceptor) was preserved. In some derivatives, the carbonyl forming the IHB could utilize the other lone electron pair to create another HB with Arg142 sidechain (and this is the Arg interacting with Pro carboxylate). When induced-fit docking was used, we also observed the interaction of the ligands to the bridging water. The binding mode was also consistent with adenosine binding (see Appendix A to compare the pose of compound **15** with the original adenosine ligand). This pose was used to create the system for the first molecular dynamics.

Compounds with monomethylated carboxamide at C-2 of the pyrazine core (**33**–**35**), using their substituent R^1^ from the most active compounds from the free unsubstituted carboxamide series, were devoid of any significant antimycobacterial activity. However, in our docking experiment, they were fully able to use the expected binding mode with similar docking scores. Obviously, the simple docking protocol did not provide sufficient background to explain the inactivity of *N-*monomethylated compounds. Therefore, we employed molecular dynamics to study the stability and overall behavior of systems with compound **15** (free carboxamide) vs its *N*-monomethylated derivative **34**.

#### 3.4.2. Molecular Dynamics

The docking pose of compound **15** (Figure 3) was used as a starting system for molecular dynamics (MD) simulations. The system was solvated, minimized, heated, and equilibrated. Three independent production 50 ns runs were performed in the NPT ensemble at 101 kPa and 300 K. The simulations were performed using NAMD under AMBER10:EHT force field, using periodic boundary conditions. For other methodological details and MD protocols, see the Appendix A. The solvated system with **15** was used as a starting point and the ligand was alchemically transformed to the *N*-methylcarboxamide derivative (**34**). One clashing water molecule was removed and the system was prepared and simulated as above.

As a result, the binding mode of compound **15** was proven to be stable in all three independent 50 ns production runs. In addition to the results of the docking, the MD simulation revealed two other important interactions-see Figure 3 for a representative snapshot of MD. The first new interaction was Glu144 sidechain carboxyl (acceptor) to hydrogen of the C-2 carboxamidic group. This led to full utilization of the HB-donating capabilities of the carboxamide, as the second hydrogen is involved in the prototypical interaction to Ala154 backbone. The HB to Glu144 was the fourth most abundant interaction, following the interactions to Ala154 backbone and interaction with the bridging water (see Table 6). Mostly, this water molecule was stabilized by four hydrogen bonds in total to ligand and receptor. On the contrary, the HB to Arg142 sidechain identified in docking turned out to be less important in MD, with significant abundance only in replica 1 of compound **15**. After visual inspection of the trajectories, we concluded that the formation of interaction with Glu144 expels Arg142 from its original position, and therefore these two interactions mutually exclude each other.

The second new interaction identified by MD is the possible halogen bond between halogen on the benzene ring (electron-deficient sigma hole) and the carboxylate anion of Glu211 side chain. This presence of the halogen bond is rather speculative, as chlorine has only a weak ability to form halogen bonds due to its small sigma hole [19]. However, on the other side, we should bear in mind that the bonding partner is electron-rich carboxylate, a very strong nucleophile. Whether classified as halogen bond or not, the fact is that in all three replicas for compound **15** the distance between Cl and the Glu211 carboxylate quickly decreased from 8 Å to 4 Å and remained stable until the end of individual runs; see Appendix A. The importance of the halogen bond should be increased in compound **18**, which is a bromo derivative as bromine is a strong halogen-bond acceptor.

The simulation of the *N-*monomethylated analogue **34** revealed the decreased stability of its binding pose. Apparently and in contrast to **15**, no HB was detected to Glu144, because of the absence of the second free hydrogen atom on the C-2 carboxamidic group. The abundance of the other HBs was significantly lower compared to compound **15** (see Table 6) and the distance between the ligand’s chlorine and the oxygen atoms of Glu211 carboxylate remained above 6 Å (Appendix A), way beyond the threshold of a reasonably strong halogen bond. For other outputs from the trajectory analyses (protein RMSD, ligand RMSD, RMSF of ligand atoms, measurements of the possible halogen bond) see the Appendix A.

In conclusion, we believe that the performed MD simulations have elucidated why *N*-methylated derivatives could have decreased affinity to mtProRS, possibly rationalizing the absence of antimycobacterial activity. Of course, proper crystallographic studies would be needed to confirm this hypothesis.

## 4. Conclusions

This study presented new antimycobacterial compounds derived from the confirmed Adachi inhibitor (**38**) of human prolyl-tRNA synthetase (hsProRS). By simplification of its peripheral substituents, we designed and synthesized more than 50 derivatives of 3-acylaminopyrazine-2-carboxamide. Compounds with unsubstituted carboxamide at C-2 of the pyrazine core and 4’-substituted benzoyl on the C-3-amino group possessed the highest antimycobacterial activity. The most effective compounds were **4**, **15**, and **18**, having larger halogen or methyl in position 4` of the benzene ring. These compounds preserved their antimycobacterial activity on multidrug-resistant strains of *M. tuberculosis*, indicating that they have a molecular target different to commonly used antitubercular agents. Active derivatives exerted insignificant cytotoxicity on the human HepG2 cell line compared to derivative **38** (Adachi), suggesting that the selectivity to human vs mycobacterial ProRS can be tuned by simple structural modifications. Molecular docking followed by unbiased molecular dynamics of the resulting complexes rationalized the mycobacterial ProRS as a probable target of the compounds. Based on the observed structure–activity relationships, we propose title compounds to be a new class of 3-aminopyrazinamide-based antimycobacterial agents, exerting the activity through the inhibition of mtProRS. This project is the first step in the successful repurposing of inhibitors of human ProRS to inhibitors of mycobacterial ProRS with antimycobacterial activity. The design of our study was based on a phenotypic screening (determination of growth inhibition in a whole cell assay). Therefore, from the acquired data, we cannot exclude that problematic pharmacokinetic properties (low water solubility, low penetration to (myco)bacteria, metabolism, or efflux, for example) might be the reason for the inactivity of some compounds. In the future, we plan to confirm the binding of the compounds to mtProRS experimentally. Additionally, we will be looking for differences between mycobacterial and bacterial ProRS to rationalize the inactivity of the title compounds against bacteria. 

## Data Availability

The data presented in this study are available within the article and the Appendix A.

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
