# Peer review of "Adenosine-Mimicking Derivatives of 3-Aminopyrazine-2-Carboxamide: Towards Inhibitors of Prolyl-tRNA Synthetase with Antimycobacterial Activity"

_biomolecules, 2022, doi:10.3390/biom12111561_

Round 1

Reviewer 1 Report

Comments:

In this work, the authors describe the design and synthesis of derivatives of 3-aminopyrazine-2-carboxamide, several of which show promising antitubercular activity. The relationship between structure and activity is well described and the authors also propose a likely mechanism of antitubercular activity, namely inhibition of prolyl-tRNA synthetase. The latter is supported by an in silico study.

To further improve the quality of the manuscript, I would like to recommend some changes that should be made before publication of the manuscript:

- Structure-activity relationship: changes in structure obviously affect binding to the active site of the target enzyme, but the same changes also affect other properties such as solubility, permeability of the bacterial cell wall, metabolism, etc. Since the antitubercular activity of the compounds was evaluated as MIC, the above properties have a significant impact on the antitubercular activity. Thus, the reason for the low MIC of certain compounds may not necessarily be low binding energy, but perhaps low permeability of the bacterial cell wall. What is the author's opinion on this issue?
- Line 227: I would suggest replacing the word "receptor" with the word "enzyme".

- Lines 491 and 493: use of two different fonts, please correct.
- The authors should unify the spaces between numbers and units, sometimes they use spaces, sometimes not. Also, some Latin words are italicised (e.g., in silico) while others are not (e.g., in vitro); I would suggest unifying the style.

Overall, I enjoyed reading the manuscript and recommend its publication, taking into account the comments listed above.

Author Response

In this work, the authors describe the design and synthesis of derivatives of 3-aminopyrazine-2-carboxamide, several of which show promising antitubercular activity. The relationship between structure and activity is well described and the authors also propose a likely mechanism of antitubercular activity, namely inhibition of prolyl-tRNA synthetase. The latter is supported by an in silico study.

To further improve the quality of the manuscript, I would like to recommend some changes that should be made before publication of the manuscript:

- Structure-activity relationship: changes in structure obviously affect binding to the active site of the target enzyme, but the same changes also affect other properties such as solubility, permeability of the bacterial cell wall, metabolism, etc. Since the antitubercular activity of the compounds was evaluated as MIC, the above properties have a significant impact on the antitubercular activity. Thus, the reason for the low MIC of certain compounds may not necessarily be low binding energy, but perhaps low permeability of the bacterial cell wall. What is the author's opinion on this issue?

RESPONSE: We agree. We used a phenotypic screening (measurement of growth inhibition activity on whole cells) for two reasons. The first one is apparent – we still do not have the enzyme at our hands (planned for future experiments). The second reason is that by this approach, we get directly compounds which are potent inhibitors of mycobacterial growth. In the vast majority of the SAR discussions in the manuscript, we discussed growth inhibition activity. However, the reviewer is correct that, in several places, we made implications towards possible binding (inhibition) of the enzyme (mtProRS). To make a clear statement, we added a sentence to Conclusion, reading:

“The design of our study was based on a phenotypic screening (determination of growth inhibition in a whole cell assay). Therefore, from the acquired data, we cannot exclude that problematic pharmacokinetic properties (low water solubility, low penetration to (myco)bacteria, metabolism, or efflux, for example) might be the reason for the inactivity of some compounds.”

We, for example, do not think water solubility was the issue because the most active compounds were of relatively lower solubility (as judged from the maximum concentrations reached e.g. in the HepG2 cytotoxicity assay). Since we do not have good data on pharmacokinetics (which will be done in the future), we will refrain now from further judgments and leave this issue to the opinion of the reader.

- Line 227: I would suggest replacing the word "receptor" with the word "enzyme".

RESPONSE: At this occurrence, the term “receptor” was changed to “enzyme (mtProRS)”. However, we would like to point out that in other parts of the manuscript, especially in the in silico part, we retained the term “receptor” and we use it in the meaning of “receptor for docking”. In this context, it is usual in the field to use the term “receptor” for all types of targets, including enzymes, It is not meant to mean a (membrane, nuclear) receptor as a part of a signalling pathway.

- Lines 491 and 493: use of two different fonts, please correct.

RESPONSE: Thanks for your careful reading. The error was already fixed by the editorial office when formatting the manuscript to the appropriate template.

- The authors should unify the spaces between numbers and units, sometimes they use spaces, sometimes not. Also, some Latin words are italicized (e.g., in silico) while others are not (e.g., in vitro); I would suggest unifying the style.

RESPONSE: Thanks for your careful reading. All the styles were unified. Values and units with a separating space. Latin words NOT italicized upon consultation with the editorial office (progressive typing preferred).

Overall, I enjoyed reading the manuscript and recommend its publication, taking into account the comments listed above.

Reviewer 2 Report

The manuscript describes the interation-mode and activity of potential inhibitors of mycobacterial ProRS, and the Multidrug-resistant theme was addressed.

There is interesting information in the in silico results and they should be of interest to the readers of Biomolecules. The manuscript reads well.

Minor points:
In the PDF for review, a few times, the error
Error! Reference source not found. was found, which makes difficult the reading and the obtention of respective References to consult.

Lines 46-48: Phrase with a strange break/gap.

In what regard in silico studies, the protocols for both docking and MD simulation were well conducted. Personally, I would prefer to see more information directly in the manuscript Methods, rather than in Supporting Information. For example, docking program and options, ligands preparation (design and structural optimization), and MD details.

The in silico results were well discussed as well.

Author Response

The manuscript describes the interaction-mode and activity of potential inhibitors of mycobacterial ProRS, and the Multidrug-resistant theme was addressed.

There is interesting information in the in silico results and they should be of interest to the readers of Biomolecules. The manuscript reads well.

 Minor points:In the PDF for review, a few times, the error Error! Reference source not found. was found, which makes difficult the reading and the obtention of respective References to consult.

 RESPONSE: Sorry for difficulties in reading. Those errors were caused by malfunctional cross-referencing in the Word document. All errors were corrected and the references to Figures, Schemes and Tables were created manually.

Lines 46-48: Phrase with a strange break/gap.

 RESPONSE: Corrected.  

In what regard in silico studies, the protocols for both docking and MD simulation were well conducted. Personally, I would prefer to see more information directly in the manuscript Methods, rather than in Supporting Information. For example, docking program and options, ligands preparation (design and structural optimization), and MD details.

 RESPONSE: As suggested, the experimental details and protocols were moved from supplementary file to the main text. We agree that the in silico methods are one of the cornerstones of this manuscript, therefore they deserve to be more visible.

The in silico results were well discussed as well.